# Liposome Formulations for the Strategic Delivery of PARP1 Inhibitors: Development and Optimization

**DOI:** 10.3390/nano13101613

**Published:** 2023-05-11

**Authors:** Carlota J. F. Conceição, Elin Moe, Paulo A. Ribeiro, Maria Raposo

**Affiliations:** 1CEFITEC, Department of Physics, NOVA School of Science and Technology, Universidade NOVA de Lisboa, 2829-516 Caparica, Portugal; cj.conceicao@campus.fct.unl.pt; 2Laboratory of Instrumentation, Biomedical Engineering and Radiation Physics (LIBPhys-UNL), Department of Physics, NOVA School of Science and Technology, Universidade NOVA de Lisboa, 2829-516 Caparica, Portugal; pfr@fct.unl.pt; 3Institute of Chemical and Biological Technology (ITQB NOVA), The New University of Lisbon, 2780-157 Oeiras, Portugal; elinmoe@itqb.unl.pt; 4Department of Chemistry, UiT—The Arctic University of Norway, N-9037 Tromsø, Norway

**Keywords:** PARP1 inhibitors, Veliparib, Rucaparib, Niraparib, liposomes, cancer therapy

## Abstract

The development of a lipid nano-delivery system was attempted for three specific poly (ADP-ribose) polymerase 1 (PARP1) inhibitors: Veliparib, Rucaparib, and Niraparib. Simple lipid and dual lipid formulations with 1,2-dipalmitoyl-sn-glycero-3-phospho-rac-(1′-glycerol) sodium salt (DPPG) and 1,2-dipalmitoyl-sn-glycero-3-phosphocoline (DPPC) were developed and tested following the thin-film method. DPPG-encapsulating inhibitors presented the best fit in terms of encapsulation efficiency (>40%, translates into concentrations as high as 100 µM), zeta potential values (below −30 mV), and population distribution (single population profile). The particle size of the main population of interest was ~130 nm in diameter. Kinetic release studies showed that DPPG-encapsulating PARP1 inhibitors present slower drug release rates than liposome control samples, and complex drug release mechanisms were identified. DPPG + Veliparib/Niraparib presented a combination of diffusion-controlled and non-Fickian diffusion, while anomalous and super case II transport was verified for DPPG + Rucaparib. Spectroscopic analysis revealed that PARP1 inhibitors interact with the DPPG lipid membrane, promoting membrane water displacement from hydration centers. A preferential membrane interaction with lipid carbonyl groups was observed through hydrogen bonding, where the inhibitors’ protonated amine groups may be the major players in the PARP1 inhibitor encapsulation mode.

## 1. Introduction

Cancer, a multifactorial disease, has a standardized treatment strategy that resorts to the use of radiation therapy coupled with surgical intervention and chemotherapeutic agents [1]. Cytotoxic side effects from ionizing radiation on normal cells are also reported, which affect biological molecules directly through altered DNA moieties, and indirectly via the generation of reactive oxygen species (ROS) [2,3]. 

The efficiency of radiation therapy can be maximized with the use of molecules that can modulate radiation effects, among which DNA repair inhibitors are included [4,5]. These molecules have specific cellular targets that are involved in signaling pathways related to homologous repair (HR), base excision repair (BER), and non-homologous end joining (NHEJ), such as poly (ADP-ribose) polymerase-1 (PARP1) [1,4,6,7]. PARP1 inhibitors, such as Veliparib, Niraparib, and Rucaparib, were developed to specifically interact with the protein’s catalytic domain and prevent PARP1’s enzymatic activity. In this way, PARP1’s biological cycle and DNA repair process are stalled, undermining cellular viability [4,5]. However, some drawbacks are associated with this therapy, such as the cytotoxic effect on normal cells, high compound clearance rates in the organism, and drug interaction with plasma proteins. Values as high as 83% binding with human plasma protein have been reported for Niraparib after entering the blood through the digestive system [8]. 

Thus, a liposome-based delivery system may be the answer to surpassing these downsides and increasing therapeutic efficacy and efficiency [9]. Liposomes have been used and investigated as efficient transport systems and as a means to achieve target delivery of therapeutic agents [9,10,11,12,13]. This ability is mostly due to its biocompatibility, biodegradability, low toxicity, lack of immune system activation, and incorporation capability for both hydrophilic and hydrophobic drugs [9,10,11,12,13]. Furthermore, liposomes can act as a protective barrier for photosensitive molecules, protecting the drug’s structural molecular integrity and therapeutic activity [14,15]. Additionally, lipidic nanoparticles can be employed as a biomimetic system for lipid–drug interaction studies, as a means to reduce drug cytotoxicity, and for the development of co-loading systems to attain a high degree of drug synergistic therapeutic effect [9]. 

In this work, the development of a lipid nano-delivery system was attempted for three specific PARP1 inhibitors: Veliparib, Rucaparib, and Niraparib. Simple lipid formulations were tested and developed following the thin-film method, coupled with two sonication methods (bath and tip sonicators). Liposome production, optimization, and compound encapsulation were extensively explained and discussed.

## 2. Materials and Methods

### 2.1. Chemicals

1,2-Dipalmitoyl-sn-glycero-3-phospho-rac-(1′-glycerol) sodium salt (DPPG) (MW 744.96 g/mol) and 1,2-dipalmitoyl-sn-glycero-3-phosphocoline (DPPC) (MW 734.05 g/mol) were purchased from AvantiPolar Lipids (Alabaster, AL, USA) and CordenPharma (Plankstadt, Baden-Wurttemberg, Germany), respectively. PARP1 inhibitors Veliparib (MW 244.3 g/mol), Rucaparib phosphate (MW 421.4 g/mol), and Niraparib (MW 320.4 g/mol) were purchased from AdooQ^®^ Bioscience (Irvine, CA, USA). 

### 2.2. Liposome Preparation 

DPPG and DPPC liposomes were prepared using the dry thin-film method [16,17,18], where 1 mM of lipid was dissolved in a chloroform and methanol 4:1 (*v*/*v*) mixture. The solvents were evaporated using a gentle stream of nitrogen. To remove residual solvent traces, samples were left overnight in a desiccator, under vacuum. Lipid films were then hydrated at 47 °C for 2 h with Milli-Q ultrapure water (Millipore, Burlington, MA, USA) [16,17,18] to a final concentration of 1 mM. Bath and tip sonicators (UP200S (200 W, 24 kHz), Hielscher Ultrasonics, GmbH, Teltow, Germany) were used to sonicate vesicle suspensions to obtain small unilamellar vesicles (SUV) [16,17,18]. Different time intervals and cycles were tested: 1–3 cycles of 5 min and 15–25 cycles of 30 s with 1 min intervals between sonication cycles. Veliparib, Rucaparib, and Niraparib were added to the DPPG formulations at different final concentrations of 20, 100, and 200 µM during the lipid dissolution in the organic solvent mixture (organic phase) and the hydration (aqueous phase) steps of liposome preparation. For simplification, we will refer to the drug addition during the two steps of liposome preparation as organic phase supplementation and aqueous phase supplementation, respectively. Non-trapped inhibitor molecules were removed from liposomes by dialysis (Spectra/Por^®^ 4 Dry Standard RC membrane, MWCO 12–14 kDa, Spectrum Labs, Biotech, San Francisco, CA, USA) for 48 h at 4 °C. The concentration of PARP1 inhibitors encapsulated in DPPG/C liposomes was determined by UV–Vis spectroscopy, by measuring absorbance at 295 nm (Veliparib), 360 nm (Rucaparib), and 311 nm (Niraparib) using a Shimadzu UV-1800 UV/Visible Scanning Spectrophotometer (Kyoto, Japan). Liposome encapsulation efficiency (EE%) and loading capacity (LC%) were determined using the following equations:(1)$$\mathrm{EE}\%=\frac{C_{encap}}{C_{total}}\times 100$$
and
(2)$$\mathrm{LC}\%=\frac{C_{encap}}{C_{lipid}}\times 100$$
where $C_{encap}$ is the concentration of inhibitor encapsulated after dialysis, $C_{total}$ is the concentration of inhibitor before dialysis, and $C_{lipid}$ is lipid concentration.

### 2.3. Spectral Measurements 

Both UV–Vis and FTIR spectra were used to analyze drug encapsulation and interaction modes with liposomes. UV–Vis spectra were acquired with a Shimadzu UV-1800 UV/Visible Scanning Spectrophotometer (Shimadzu). FTIR spectra were recorded on a Bruker IFS 66/S (Billerica, MA, USA) spectrometer in absorbance mode, with a wavenumber range of 500 to 4000 cm^−1^, a resolution of 4 cm^−1^, and 128 scans. Samples were deposited on CaF_2_ supports, and water was evaporated in a desiccator under vacuum conditions. CaF_2_ clean support spectra were subtracted from all sample spectra. 

### 2.4. Dynamic Light Scattering

Liposome size, polydispersity index, and zeta potential were determined by dynamic light scattering measurements using a Zetasizer Nano ZS (Malvern Instruments, 633 nm He-Ne laser, measuring angle θ = 173°) (Malvern, Worcestershire, United Kingdom). Size measurements were carried out using a 0.3 mm quartz cuvette (ZEN 2112) for low-volume samples, and the temperature was set to 25 °C. The analyzed liposomes were diluted to 10 µM and 200 µM concentrations for size and zeta potential analysis, respectively. At least three measurements per sample were undertaken, with a 1 min waiting time and a 30 s equilibration time. The lipid refractive index input was 1.45, and the dispersant was defined as 1.33. Zeta potential measurements were attained using the Smoluchowski equation and Malvern disposable sample cells (DTS1070). Intensity particle size distributions and zeta potential values were recorded with Zetasizer software v.7.11. Mean and standard deviation values are presented.

### 2.5. In Vitro Release of PARP1 Inhibitors

Drug release studies [19] of DPPG-encapsulating PARP1 inhibitors Veliparib, Rucaparib, and Niraparib were performed following an adaptation of Sivadasan et al. [19] using Slide-A-Lyzer™ Mini Dialysis devices (MWCO 10K) (ThermoFisher, Waltham, MA, USA). It should be noted that other Slide-A-Lyzer™ systems, namely Slide-A-Lyzer™ Cassete, like those employed here, have previously been used to build a “Liposome Dialysis” system [20]; however, the sample volumes needed are greater (250–300 µL) than the device that was used in the present work (50 µL). The liposome formulation was applied (50 µL) to the mini dialysis devices, sealed, and immersed in 1.2 mL of PBS 1x pH 7.4. The dialysis system was incubated in a Thermoblock (Eppendorf, Hamburg, Germany) at 37 °C with gentle agitation. The temperature was selected taking into consideration typical human body temperature. At fixed time intervals, samples were collected from the dialysate buffer and replaced with fresh PBS. Dialysate was analyzed by UV–Vis spectroscopy, and absorbance at 295 nm, 360 nm, and 311 nm was measured for Veliparib, Rucaparib, and Niraparib, respectively. Control (pure drug) samples were also subjected to the dialysis process, and the concentration used was equivalent to the final drug concentration encapsulated in the liposome formulations. All formulations were analyzed in triplicate, and cumulative drug release percentages were determined.

### 2.6. Release Kinetic Models

Data obtained from in vitro drug release analyses were plotted using various kinetic models with the KinetDS software [21,22] to study drug release kinetics from DPPG. The best model criteria employed were empirical R^2^:(3)$$R_{emp}^{2}=1-\frac{\sum_{i=1}^{N}{\left(y_{i}-{\hat{y}}_{i}\right)}^{2}}{\sum_{i=1}^{N}{\left(y_{i}-y_{AV}\right)}^{2}}$$
and the Akaike information criterion (AIC) defined by
(4)$$\mathrm{AIC}=2K+N\cdot [\ln\left(\sum_{i=1}^{N}{\left(y_{i}-{\hat{y}}_{i}\right)}^{2}\right)]$$
where $y_{i}$ is the observed value, ${\hat{y}}_{i}$ is the model-predicted value, and $y_{AV}$ is the average output value.

The mathematical models used that presented best fit were Weibull:(5)$$Q=1-exp\left[\frac{-{\left(t\right)}^{b}}{a}\right];$$
modified Weibull with Lag Time (*T_LAG_*):(6)$$Q=1-exp\left[\frac{-{\left(t-T_{LAG}\right)}^{b}}{a}\right];$$ Michaelis–Menten:(7)$$Q=\frac{Q_{max}\cdot t}{k+t};$$ Michaelis–Menten (*T_LAG_*):(8)$$Q=\frac{Q_{max}\cdot \left(t-T_{LAG}\right)}{k+\left(t-T_{LAG}\right)};$$ and Hixson–Crowell:(9)$$Q^{\frac{1}{3}}=k\cdot t+Q_{0}^{\frac{1}{3}}$$
where Q is the amount (%) of drug released at time t, $Q_{o}$ is the start value of Q, $Q_{max}$ is the maximum value of Q (100%), t is time, $T_{LAG}$ is lag time, and a and b are constants. 

To determine the drug release mechanism, the obtained data were plotted using the Krosmeyer–Peppas equation: (10)$$Q=k\cdot t^{n}$$
where n (release exponent) was used as the drug release mechanism determinant parameter.

## 3. Results and Discussion

Lipidic formulation production, optimization, and compound encapsulation will be extensively explained and discussed in the following subsections. 

### 3.1. Development of the PARP1i Lipidic Nano-Delivery System

The preparation of DPPG and DPPC single-lipid formulations followed the procedure described in Section 2. Once this procedure was completed, the sonication was optimized to reduce the particle size to a desired range.

#### 3.1.1. Optimization of the Sonication Process

To optimize DPPG and DPPC liposome size, two sonication methods were used (bath-type and tip sonicators), and several time intervals and cycles were employed (1–3 cycles of 5 min and 15/20/25 cycles of 30 s). The bath-type sonicator produced unstable formulations in all conditions, while the tip sonicator was able to produce relatively stable populations. This may be related to the cavitation intensity and ultrasonic wave distribution, where the bath sonicator tends to have an uneven ultrasonic irradiation distribution and the tip sonicator tends to have a greater localized effect that translates into a higher intensity and efficiency in the process [23,24]. Cycles of 5 min revealed a non-homogeneous and unstable liposome population, with the occurrence of particle flocculation for both types. The best results were obtained with a reduced time (30 s cycles) and the tip sonicator (UP200S, 200 W, 24 kHz), since long sonication periods tend to hinder liposome stability through the promotion of lipid oxidation associated with elevated cavitation exposure [25]. Cycles of 30 s (15, 20, and 25) produced stable DPPC and DPPG liposomes, with the presence of two particle populations of small (<95 nm) and large (>0.5 µm) sizes (Figure 1 and Table A1 and Table A2). The DPPG main population presented a high degree of size stabilization (approx. 55 nm) regardless of the number of cycles employed (Figure 1b), while DPPC showed a constant decrease in size value dependent on the number of cycles. This could be explained by the impact of the lipids’ head-sites that govern hydrogen bonding, electrostatic interactions, and overall membrane mechanical stability [26]. Even though simulation data show that DPPC and DPPG present similar area per lipid values (62.0 Å^2^ and 67.0 Å^2^, respectively), their overall mechanical stability is quite distinct [26,27]. AFM analysis of the DPPG bilayer reported a mechanical stability of ~66 nM and that of DPPC had an intermediate value of ~19 nM [26]. The observed enhanced stabilization of the DPPG bilayers has been suggested to result from lipid coordination or is due to the effect of Na^+^ counterions adsorbed in the membrane that reduce the repulsive effect of the negative charge in the headgroup [26], thus explaining why DPPG liposome particle size appears to be more stable than DPPC when subjected to a mechanical force/stimulus, such as ultrasound.

Considering the population number (one or more populations), size, and sonication time, the 20-cycle condition was selected for liposome preparation. This sonication cycle seemed to be the best compromise between population size distribution and time of exposure to cavitation. Under these conditions, DPPG and DPPC presented z-average values of 56.2 ± 0.4 nm and 71.0 ± 0.5 nm, respectively, and polydispersity index (PdI) values below 0.3 (Figure 1c and Table 1). Regarding the main population of interest (Pop.1—population 1), a difference of approximately 19 nm in diameter is observed, as can be seen from the data displayed in Table 1. The small discrepancy could be explained by lipid interaction forces and the non-specificity of the sonication technique, which results in a broader particle size distribution. This is also visualized in the main population peak broadening, where diameter values vary from 28 to 200 nm and 20 to 170 nm, respectively, for DPPC and DPPG (Figure 1c). However, the size distribution in both formulations coincides within the range of 28 to 170 nm, showing that a significant number of nanoparticles in the DPPC and DPPG main populations are similar. The presence of large size secondary populations also has an impact on size determination and distribution, influencing scattering events and scattering intensities. DPPG presents a second population with diameters above 0.5 µM and DPPC above 2.5 µM, as can be seen in the particle size distribution of Figure 1c.

#### 3.1.2. Optimization of PARP1 Inhibitor Veliparib Encapsulation in DPPC and DPPG Liposomes

Following the selection of the best DPPC and DPPG sonication conditions, encapsulation of the PARP1 inhibitor Veliparib was attempted using 100 µM of this inhibitor. The compound was added during the lipid dissolution in the organic solvent mixture step of liposome production (organic phase supplementation). Encapsulation of Veliparib (ε_295 nm_ = 8316.7 M^−1^ cm^−1^) in the DPPG liposomes was observed with an efficiency (EE) of 49 ± 2% (Table A3), while encapsulation in DPPC was unsuccessful. In fact, extreme liposome destabilization and flocculation starting in the hydration step were observed in DPPC samples. This could be related to charge compensation given by positively charged inhibitors, which, in turn, destabilized lipid bilayer organization. However, the negative charge of the DPPG molecule was able to maintain lipid bilayer stability. Through the zeta potential determination of DPPC and DPPG without inhibitors, this event can be further explained, and the values retrieved were −17.9 ± 0.9 mV and −43 ± 2 mV, respectively, for DPPC and DPPG (Table 1). These results are a clear indication of DPPC liposome instability prior to any type of drug encapsulation and the elevated stability of DPPG (<−30 mV). The analysis of DPPG + Veliparib samples presented a zeta potential of −41 ± 2 mV, showing that inhibitor encapsulation led to an increase in liposome overall surface charge, confirming our previous theory.

Because of the very low lipid capacity (LC) attained in the DPPG + Veliparib formulation (LC = 3.9 ± 0.2%), displayed in Table A3, different conditions were tested to increase EE and LC, thus further maximizing the drug load. For this, concentration and supplementation phases were varied. Concentrations of 20, 100, and 200 μM were assessed by organic phase supplementation, and an increase in EE and LC values is verified in Table 2. It was observed that lipid capacity increases exponentially up to 10% with 200 μM of Veliparib. EE values vary inversely with drug concentration. However, a stagnation is verified for this parameter around 50–55% and occurs above the 100 µM supplementation, which indicates possible system saturation. Even though the highest concentration tested (200 µM) is far below the equimolar drug/lipid ratio, the saturation event can be related to the DMSO molecules. DMSO is the sole solvent used for the preparation of PARP1 inhibitor stocks, and it is known to interact with lipid membranes. DMSO is regularly used in cell cultures as a cryoprotectant, increasing membrane porosity and preventing water crystal formation through the promotion of vitrification at low temperatures [28]. This solvent is also known to be capable of interacting with lipids’ headgroups, leading to water displacement. Neutron diffraction analysis has shown that the DMSO sulfur atom is attracted to LUVs’ and MLVs’ negatively charged phosphate groups, while DMSO methyl groups are capable of remaining inserted in lipids’ head/pocket, which includes a phosphorus and four oxygen atoms. DMSO in PC lipids also presents a second interaction mechanism that is governed by DMSO oxygen attractive forces toward positively charged choline nitrogen moieties [29]. The water displacement mechanisms reported coincide with the DPPG (first mechanism) and DPPC (both mechanisms) expected lipophilic drug interaction regions, which clearly demonstrate the profound effect that DMSO could have on lipophilic drug encapsulation efficiency.

Further DLS characterization showed that lipid size increases with Veliparib concentration (Figure 1d), and z-average values vary from approximately 60 to 90 nm, as seen in Table 3. The population and size distribution profile, shown in Figure 1d, appears to change with drug concentration. Condition 1 (20 µM) presents two populations and a distribution profile similar to the DPPG control condition without Veliparib, as shown in Figure 1c. The increase in drug concentration led to a change in the size distribution profile, the elimination of large size populations, and the main population shift to higher diameter values (Figure 1d). PdI reduces with drug concentration (Table 3) and is intrinsically associated with a more homogeneous sample and single population profile. The population of interest (Pop.1) shows a clear shift in size from the control sample (DPPG) (56 ± 2 nm) to encapsulation samples (>56 nm) (Table 3), where the greatest differences are observed at 200 µM supplementation (134 ± 6 nm). In terms of the liposome’s surface charge, a decrease in value is observed dependent on drug concentration. Here, the values are not as straightforward, showing that 100 and 200 µM drug supplementation results in a similar shift to −40 mV, while the 20 µM condition presents a greater shift to −31 mV (Table 3). This may be explained by the presence of DMSO in the lipid membrane and the competition that occurs between Veliparib and DMSO molecules in the lipid’s headgroup space. During the dialysis process, this competition event will be a determinant. Since Veliparib stocks are prepared in 100% (*v*/*v*) DMSO solvent, drug supplementation dependent on concentration leads to positive feedback between DMSO and Veliparib final concentrations. This may explain why 100% drug encapsulation takes place with 20 µM supplementation, where the system is not saturated and the DMSO percentage is reduced (~0.02% *v*/*v*), thus allowing complete drug encapsulation. In the remaining concentrations, 100 µM and 200 µM, DMSO values were at 0.1 and 0.2%, respectively, which explains the reduction and apparent stagnation in EE (<55%) through competitive events in higher inhibitor concentrations. These results show that an increase in DMSO to values of ~0.1–0.2% fixes EE to approximately half the drug concentration input. By converting EE values to effective Veliparib concentration in DPPG membranes, it was possible to encapsulate approximately 48 µM and 100 µM in conditions 2 and 3, respectively. It was also interesting to verify that the same conditions appear to result in higher liposome stabilization with the determination of lower zeta potential values, as seen in Table 3. 

Finally, it was also verified that after dialysis, the final DMSO percentage in the liposome stock formulations, 20 µM, 100 µM, and 200 µM supplementation, was in the range of 0.02–0.05%, thus making the solvent content within the biocompatibility limit (0.05%) for future cellular studies [30]. Moreover, this biocompatibility will be further increased by the reduction of DMSO effective content upon liposome stock dilution when testing in vitro and in model organisms.

In summary, particle size is directly influenced by drug concentration, and a homogeneous liposome population is acquired at 200 µM drug supplementation. EE values are inversely related to initial drug-supplemented concentration; however, effective drug content in DPPG goes up to 100 µM of Veliparib. In terms of system stability, higher drug concentrations appear to result in a better fit, with zeta potential values around −40 mV. DMSO appears to be a competitor for drug encapsulation, and EE values appear to stagnate at 0.1–0.2% *v*/*v* DMSO.

Taking this into consideration, DPPG supplementation with 200 μM Veliparib appears to be a good and stable system to employ in any type of test and system characterization for future applications.

Additionally, in an attempt to increase LC values, phase supplementation was varied. Since PARP1 inhibitors are insoluble in water (manufacturer recommendation), initial drug supplementation was performed in the organic phase, and afterwards, simultaneous organic and aqueous phase supplementation was tested. The 200 µM Veliparib concentration condition was used and divided into dual supplementation (100 µM + 100 µM). Results showed a clear difference in EE values between conditions, where dual supplementation achieved an efficiency of 31 ± 4% (Table A4) when compared to the organic phase condition (55 ± 1%); see Table 2. A reduction of 5% in lipid capacity (LC) is observed from the organic phase to the dual phase (Table A4), which, interestingly, is very similar to the 100 µM organic supplementation condition. Additionally, the dual phase EE value appears to be even lower than the 100 µM condition, but if we take into consideration the errors associated with these values, the differences do not appear to be that significant. The apparent coincidences lead to an indication for Veliparib’s exclusive lipid bilayer encapsulation, which can be expected of a water-insoluble drug. On the other hand, the dual phase z-average and main population size values (Table A5) are similar to the 200 µM organic condition. However, the presence of a large size secondary population is consistent with the population profile from the 100 µM condition. This could be an indication that, even though the Veliparib dual system may be close to the 100 µM organic condition in EE and LC parameters, the increase in size of the main population could be related to DMSO integration on the lipid bilayer. Because DMSO addition to the lipid solution was of 0.2% *v*/*v* in the 200 µM condition, this double difference for the 100 µM condition can explain the impact on liposome swelling. Previous studies have shown that DMSO has a profound effect on the lipid membrane, and a correlation was established between the DMSO mole fraction (DMSO/water mixture) and membrane behavior [29,31]. At molar fractions below 0.1 (X_DMSO_ < 0.1), DMSO does not appear to change the membrane structure or headgroup mobility, even though water displacement is verified by the competition and gain of lipid headgroups [28]. At these ratios, the lipid gel-to-fluid transition temperature increases, thus stabilizing the gel phase through changes in the surface hydration on the lipid headgroup [29]. Thermostability in this phase can be extended by DMSO up to a 5 °C increase [31]. DMSO is, thus, capable of altering the membrane’s mechanical properties in a concentration-dependent manner. However, the DMSO effect is also variable, depending on the membrane composition and structure, which, in turn, affects membrane biological functions [31]. Infrared studies revealed that DMSO reduces the binary bilayers’ free volume, while not affecting more complex liposome systems, specifically ones involving cholesterol. In POPC/ESM 1:1 GUVs systems, DMSO noticeably affected vesicle biophysical characteristics through a “membrane loosening” event, resulting from the increase in excess surface area in the vesicle. Again, this effect was not detected in complex liposome systems involving cholesterol (POPC/ESM/chol 1:1:3) [31]. The reported data reinforce our hypothesis of DMSO affecting DPPG + Veliparib encapsulation efficiency and vesicle size through interaction with the lipid headgroup and subsequent DMSO/Veliparib competition for the negatively charged phosphate groups.

The polydispersity quality evaluation for size distribution was 0.32 ± 0.02, which is an acceptable value for this type of sample. Zeta potential analysis of dual phase supplementation shows similar values to 100 and 200 µM organic phase, with an indication of −42 ± 3 (Table A5). Making this a very stable system. In sum, organic supplementation retrieved the best results in terms of EE, LC, and sample homogeneity for Veliparib DPPG encapsulation. 

Taking all this into consideration, the thin film encapsulation protocol of PARP1 inhibitors in DPPG should follow a 200 µM drug supplementation in the organic phase.

#### 3.1.3. Ascertaining PARP1 Inhibitor Encapsulation in Single and Bi-Lipid Formulations

Encapsulation of PARP1 inhibitors Rucaparib (ε_360 nm_ = 14,144.4 M^−1^ cm^−1^) and Niraparib (ε_311 nm_ = 13,200 M^−1^ cm^−1^) was performed following the conditions previously optimized for Veliparib. DPPG drug encapsulation quality parameters retrieved slightly reduced EE values for 200 µM supplementation when compared to Veliparib values, as displayed in Table 4. A decrease of 5.8% and 4.3% for Rucaparib and Niraparib is observed, respectively. LC values are also reduced by 1% when compared to Veliparib. Both parameters are similar between Rucaparib and Niraparib. Observed differences may be due to the molecular structure of the compound, the protonation state, the overall charge, and spatial hindrances based on the inhibitor’s molecular size. Theoretical topological surface areas show an increase in calculated values in the following order: Niraparib (72.9 Å^2^) < Veliparib (83.8 Å^2^) < Rucaparib (135 Å^2^) [32,33,34]. Even though differences between Niraparib and Veliparib do not follow the expected tendency (Veliparib < Niraparib), we must consider that this parameter is calculated taking into consideration only the N and O atoms, which have the same representativity in both molecules (five in total), as shown in Table 4. However, when comparing overall molecular structures, it is evident that the total atom composition is greater in Niraparib than Veliparib, which, in turn, translates into a greater atomic mass. The design of PARP1 inhibitors targeting the protein catalytic domain follows specific guidelines. They all have in common a benzamide moiety, while the remaining structure is adjustable and designed to best fit the catalytic hydrophobic pocket, which thus ensures high target specificity [5,35,36]. These compounds are designed to mimic NAD+ molecules (specifically the nicotinamide core structure) that are essential for PARP1’s transferase activity [5,36]. 

Nevertheless, for all three inhibitors, the determined EE values are exceptionally good (≥50%) and translate into an effective drug cargo of 90 to 100 µM in DPPG liposomes.

Since DPPC drug encapsulation was not successful, the development of a complementary liposome system was attempted. To ensure the presence of PC lipids and further increase biological compatibility in the liposome formulation, we developed an equimolar DPPG/DPPC complex following the protocol previously described. The bi-lipid liposome formulation was relatively stable (<−30 mV), presenting a surface area charge of −38.8 ± 0.7 mV. In comparison to DPPG control samples (see Table 1), the best stability results were observed in the single liposome formulation. The decreased stabilization of DPPG/DPPC can be expected if one considers the overall DPPC lipid charge and determined zeta potential values (−17.9 mV). DPPG/DPPC 1:1 size distribution analysis revealed a single population with a mean diameter size of 98.0 ± 1.76 nm, a z-average value of 77.0 ± 0.8 nm, and a PdI value of 0.22 ± 0.01. In comparison with DPPC and DPPG single-lipid formulations (Table 1), 20 sonication cycles resulted in an overall increment in size values of the complex liposome formulation. This can be related to the attractive and repulsive forces between DPPC and DPPG headgroups, as well as the Na^+^ counterion effect on the equilibrium of these forces.

Since the DPPG/DPPC 1:1 formulation presented a diminished degree of stabilization compared to PG liposomes, a 100 μM drug supplementation was employed. EE and LC values for the three inhibitors were low, which was expected. EE values were around 40% (see Table 4), and close to the 100 µM Veliparib supplementation in DPPG liposomes (Table 2). Still, some differences are observed, with a decrease in efficiency that ranges from 9 to 15%. Among PARP1 inhibitors, differences are also verified, with Rucaparib presenting the best encapsulation values and Niraparib presenting the lowest EE value (33.7 ± 0.9%). The differences in encapsulation efficiencies can be explained by the lipid formulation and the lower presence of DPPG lipids in the DPPG/DPPC liposome formulation. Additionally, liposome and inhibitor overall charge will be determinants of lipid–inhibitor interaction. Lipid susceptibility to DMSO presence can also explain EE differences between liposome formulations, with lower mechanical stabilization of DPPC over DPPG lipid [26]. LC values are like those observed with 100 μM supplementation in DPPG liposomes. These results are a clear indication that the DPPG/DPPC formulation has similar behavior to the DPPG 100 μM drug supplementation condition but does not present the same good EE values.

In the following section, liposome formulations with encapsulated PARP1 inhibitors will be extensively characterized and discussed. Additionally, a quality assessment of each formulation as a potential therapeutic alternative will be performed, and an effective lipid–inhibitor interaction will be determined. 

### 3.2. Lipid Delivery System Characterization—Quality Assessment

DLS zeta potential characterization has shown that in terms of surface charge, DPPG-encapsulating inhibitor liposomes have a better fit than the bi-lipid formulation, with values achieved being far below the −30 mV stability boundary. Binary DPPG/DPPC liposome-encapsulating inhibitors presented values over −30 mV (Table 5), stating that the formulations are not as stable as would be desired. In terms of size distribution, DPPG and the bi-lipid formulation present similar z-average values around 90 to 100 nm. All formulations, except DPPG/DPPC (1:1) + Rucaparib, present only one population with a diameter above 100 nm, as seen in Table 5. The smallest particle size is verified in the Veliparib formulation, which would be expected if the molecular structure and topological surface areas of the inhibitors were taken into consideration, as discussed previously. Additionally, from the analysis of published PARP1-inhibitor crystal structures, it is possible to get an idea of each inhibitor’s 2D dimensions [37,38]. This allows us to determine reference values for putative maximum lengths and widths of Veliparib, Niraparib, and Rucaparib using the RSCB PDB data treatment software [39,40,41,42]. The inhibitors’ maximum values follow the tendency of Rucaparib > Niraparib > Veliparib for width, and Niraparib > Rucaparib > Veliparib for length (Figure A1). These data are an indication of the putative spatial constrictions and in which order they may occur. 

Overall, DPPG liposomes appear to be the best formulation for PARP1 inhibitor encapsulation, presenting good EE and LC values, as well as a high degree of stabilization (zeta potential < −30 mV). In terms of liposome size distribution, a single population profile is attained, and particle diameter values are somewhat coherent. Main population (Pop.1) particle size contribution is defined as 130 ± 3 nm, 135 ± 10 nm, and 132 ± 7 nm for each inhibitor (Table 5). Between the three formulations, size distribution appears to be more homogeneous in DPPG + Veliparib liposome particles (Figure A2). 

Since DPPG simple formulation presented the best fit for further exploitations as PARP1 inhibitor delivery system, an evaluation over time was performed concerning particle size stability along a 28-day time period. Samples were stored at 4 °C after production, with Z-average and main population size values taken into consideration and analysed in parallel, seen in Figure 2. The DPPG control sample (without inhibitor) showed an initial increase in size values after 7 days of storage, with a return to approximately initial size values after 14 days, as it can be seen from Figure 2a. After 28 days of storage, liposome samples have shown a stabilization of the size of the main population, but z-average values increased significantly with detection of large size particles (>0.5 µm). This was a clear indication of sample destabilization after approximately one month storage.

The observation of size variation of DPPG liposomes with time is not uncommon for simple liposome formulations. It has already been reported for DMPC, DPPC, and DSPC simple formulations that there are increases and decreases in liposome size distribution [43]. In these, particle size stabilization was enhanced and achieved with the introduction of cholesterol to the liposome formulations. Stabilization was due to increases in transition temperatures and membrane elasticity range, thus improving lipid organization, packing, and formulation robustness [43]. 

The liposome formulation containing PARP1 inhibitors revealed specific size variation patterns for each compound, different from the control. Veliparib appears to stabilize DPPG liposome size until 14 days of storage, for both z-average and main population values, which is an indication that the liposome product/formulation retained the same size characteristics as those observed after the production stage. Over 14 days, size distribution values show an increase in particle diameter with the surfacing of large size populations (Figure 2b). On the other hand, Rucaparib appears to have a profound effect on particle size distribution, with z-average and main population values becoming closer to each other (Figure 2c). This may be an indication that Rucaparib increases sample homogeneity in terms of size population distribution. After 14 days of storage, it was verified that there was an increase in z-average values, while the main population size was maintained. This was related to the appearance of large size populations due to particle aggregation. The DPPG + Niraparib formulation, on the other hand, shows an inverted tendency from the DPPG control sample. Variance is observed in the main population size, while z-average size variation is more stable. The main population shows a decrease in particle size until 7 days of storage, which is followed by an increase until 14 days. Past this time, the particle size increase rate is slowed down. From a physical sample evaluation, it was observed that until 14 days of storage, all samples were translucid with no apparent deposit; however, after approximately one month, this event was observed. This indicates that after one-month, DPPG liposome formulations present aggregation events, with large size particle destabilization. Nevertheless, for all drug formulations, it was verified that particle size stabilization occurred until 14 days of storage at 4 °C, with no deposit formation/flocculation. Taking into consideration cholesterol studies on liposome stability [43,44], it can be inferred that the tested inhibitors may be affecting DPPG stability in a similar manner. By putative electrostatic interaction and hydrogen bonding between the inhibitors and lipid headgroups, DPPG stabilization is being favored through the increase of membrane elasticity temperature range (increase in transition temperature leads to a more permanent gel-phase state) and promotion of lipid organization, thus translating into a high degree of system robustness and fit.

The DPPG formulations appear to be a good option for PARP1 inhibitors’ encapsulation and future therapeutic applications. We verified that encapsulation is favored using negatively charged lipids and state that DPPG should be employed when designing a lipid carrier for these compounds. Furthermore, the produced liposome formulations can be used as a base for further liposomal surface functionalization with coatings such as polymers, polyelectrolytes, antibodies, or even conjugation with other systems [15,44,45]. This will aid in overall system stability, biocompatibility, target specificity, and controlling drug release [15,45,46].

### 3.3. Drug Release Analysis and Kinetics—Time of Circulation Determination 

The biophysical features of liposome-encapsulating lipophilic compounds are greatly affected and determined by their core–shell properties [43,44]. However, additional factors are also in play when we talk about drug release. Factors such as buffer/medium, the nanocarrier units’ physicochemical properties, and the nanocarrier’s internal structure are of high importance for release rates and the determination of suitable delivery systems [47].

An in vitro analysis was performed with the three DPPG-encapsulating PARP1 inhibitor systems and compared to the respective standard drug solutions. Each control sample was prepared at the concentration that was present in the liposome system. PBS 1x was used as the dialysis buffer, and compound concentration was determined from the dialysate. Absorbance measurements were performed at the absorbance wavelengths of 295, 360, and 311 nm for Veliparib, Rucaparib, and Niraparib, respectively. It was verified in all samples that there was an initial burst release due to the dialysis system, thus contributing a lag time to the cumulative drug release profile, as can be seen in the plots in Figure 3. Release patterns are distinct for each inhibitor, with control samples presenting after 3 h at 37 °C values of 66 ± 1%, 104 ± 1%, and 68.5 ± 0.9% for Veliparib, Rucaparib, and Niraparib, respectively. Corresponding DPPG drug release percentage values were 48 ± 9%, 96 ± 2%, and 54 ± 2%. Differences observed between control and liposome formulations show that the presence of DPPG led to an overall decrease in drug release rates. Past 7 h, cumulative drug release values were as follows: 74 ± 15%, 100.0 ± 0%, and 59 ± 11% (Figure 3), showing that DPPG + Veliparib/Niraparib are best suited for long-term release. The DPPG + Rucaparib system was efficient in delaying drug release rates until 3 h; however, after this timepoint, cumulative values converged to one of the control samples (Figure 3b).

Drug release from nano-delivery systems is determined by the physical and chemical properties of the system, including the drug and nanocarrier. Features such as porosity, surface roughness, chemical composition, molecular weight, degradation rate, particle size, compound dosage, and drug–matrix interactions are of extreme importance [48]. In our case, the variability observed between systems may be related to the entrapment efficiency of the inhibitors in the nanocarrier bilayer, to the inhibitors orientation and mode of insertion in the bilayer, as well as to the membrane’s rigidity, porosity, and gel–fluid state [44,47].

To understand drug release kinetics and determine the DPPG potential nanocarrier applicability and efficiency, the data obtained were plotted following various mathematical models. Among them, Weibull (*T_LAG_*) and Michaelis–Menten equations better explained drug release from DPPG, presenting the highest linearity with the highest value of R^2^ and the lowest value of the Akaike information criterion (AIC), as can be seen from the data displayed in Table 6. The first equation gave the best fit for Veliparib and Niraparib release data, while the second equation explains the Rucaparib system. The Hixson–Cromwell model did not explain the drug release data. Krosmeyer–Peppas n values suggest that all systems present complex drug release mechanisms. Calculated values with best fit point to a non-Fickian diffusion mechanism (0.45 < n < 0.89) (Krosmeyer–Peppas (*T_LAG_*)) for all three inhibitors [49]. Further confirmation of transport complexity is given by the Weibull β (shape) parameter, with values < 1 for Veliparib and Niraparib, and >1 for Rucaparib (Table 6).

β values ≤ 0.75 are linked to Fickian diffusion mechanisms, while values between 0.75 and 1 are associated with combined mechanisms of Fickian diffusion and controlled transport by nanoparticle swelling. A shape factor above 1 implies very complex drug release mechanisms, characterized by an initial non-linear increase in drug rate release, followed by an equal decrease event [50]. This mechanism is quite perceptible in Rucaparib’s cumulative drug release graph (Figure 3b). Thus, our results state that the Veliparib and Niraparib release mechanisms in DPPG are governed by a combined system of Fickian and non-Fickian diffusion processes (n and β values), involving nanoparticle swelling and deswelling events. Rucaparib, on the other hand, presents a complex mechanism associated with anomalous diffusion with a possible link to super case II transport, if the n > 0.89 value for the conventional Krosmeyer–Peppas model is taken into consideration (Table 6) [49]. Super case II is mainly associated with stress and a polymer state transition due to swelling in water or biological fluids [51]. The combination of anomalous case II and super case II behaviors has previously been related to hydrogel porosity [49], which can give us some indications about the DPPG-encapsulating PARP1 inhibitor release system. Taking into consideration the DPPG transition temperatures (Tc = 41 °C), the release assay temperature (Ta = 37 °C), and the presence of DMSO, it is within reason to hypothesize that while performing the release assay, liposome samples could have been in their gel-state phase (Ta < Tc), presenting some pore structures resulting from the presence of DMSO, which is in line with the anomalous and porous theory. In this way, super case II transport, indicated by n values above 0.89, and anomalous transport verified in the liposome systems can be explained. All these are due to drug release mechanisms governed by membrane swelling transport due to osmotic pressure, membrane hydration, and a protonation state [48]. 

In summary, it has been shown that DPPG-encapsulated inhibitors presented slower release rates than control samples. DPPG + Veliparib/Niraparib were considered the most suitable formulations, with complete drug release delayed until 7 h. Rucaparib formulation was efficient until 3 h of dialysis; however, after this timepoint, cumulative values converged to those from the control sample. The three inhibitors presented complex release mechanisms, with Veliparib and Niraparib being governed by a combination of diffusion-controlled (Fickian) and non-Fickian diffusion mechanisms, while anomalous and super case II transport was verified for Rucaparib. Anomalous drug transport events could be related to membrane swelling events and the presence of porous structures due to DMSO. Further testing and optimization should be performed to enhance system stability in circulation and drug release rates. This could be achieved through the addition of stabilizer components to the liposome membrane, such as cholesterol or other lipidic components. The addition of polymeric stabilizers such as PEG or polyelectrolytes may help to increase membrane stability and decrease drug release rates [15,19]. In addition, the reduction in DMSO content could help with membrane porosity; however, this will be limited and related to drug solubility and dialysis procedures. 

### 3.4. PARPi and Lipid Delivery System Biophysical Interaction—Spectroscopic Characterization

Spectroscopic characterization was performed for DPPG-encapsulating systems using UV–Vis and Fourier-transform infrared (FTIR) spectroscopies.

The UV–Vis spectra of DPPG liposomes in Figure 4 present a maximum absorption peak near 194 nm, which is consistent with a typical liposome band at 194.4 ± 0.7 nm [18,52]. This is assigned to the lone-pair transition of carbonyl oxygen to the antibonding π_CO_ valence orbital, to n_O_ → π*_CO_, or to the valence shell electronic excitation of hydroxyl groups [18]. DPPG-encapsulating PARP1 inhibitors present characteristic inhibitor maximum absorbance peaks in the 224–450 nm region (Veliparib: 270 and 295 nm; Rucaparib: 238, 281, 323, and 360 nm; Niraparib: 239 nm, 311 nm, and 342 nm), confirming drug encapsulation (Figure 4). Upon drug incorporation into the lipid carrier, it is verified that Rucaparib’s 278 nm and Niraparib’s 306 nm pure drug maximum absorbance peaks, shown as colored dash lines in the UV–Vis spectra in Figure 4, present a significant change in the peak position toward higher wavelengths, i.e., a red shift, when DPPG is encapsulating these drugs. The observed shift could be a result of a higher protonation state [53] and/or the presence of a substituent (e.g., -OH) in the chromophore [54] in the inhibitor molecules. The DPPG + Veliparib sample does not present such significant shifts in its peaks in the UV–Vis spectra. The observed red shift of the maximum absorbance peaks in the case of Rucaparib and Niraparib molecules encapsulated into liposomes serves as an additional indication of the interaction of these inhibitors with the liposome membrane and the indication of drug encapsulation. Additionally, the characteristic absorbance peak of DMSO solvent (211 ± 6 nm) is also perceived in the region around 205–210 nm, due to the n → π* electronic transition from the unshared electron pair of the oxygen atom [55].

Characteristic infrared absorption bands and assignments are displayed in Table A6 for DPPG (Figure 5) and Table A7, Table A8 and Table A9 for PARP1 inhibitors (Veliparib, Rucaparib, and Niraparib) (Figure A3).

The spectra for the DPPG formula were obtained by subtracting the CaF_2_ support without the cast film. The control sample spectra (dark line), shown in Figure 5, present typical DPPG bands with a broad peak in the region above 3000 cm^−1^, which is attributed to O-H and C-H vibrational modes, while the 2800–3000 cm^−1^ region refers to both symmetric and anti-symmetric stretching vibrations of the CH_2_ groups. Incorporation of PARP1 inhibitors is perceived in these regions with the narrowing and sharpening of the band above 3000 cm^−1^ (due to C-H and N-H absorbance contributions from inhibitor molecules) and the increased absorbance of CH_2_ vibrational bands. Band narrowing above 3000 cm^−1^ indicates possible inhibitor incorporation through the exclusion/displacement of water molecules, due to the reduction of OH vibrational modes, as well as incorporation in the bilayer membrane by peak sharpening and absorbance increases above the 2800 cm^−1^ region (Figure 5a). Furthermore, in the Figure 5a inset, an additional band is detected at 2872 cm^−1^, between liposome CH_2_ vibrational modes, consistent with the inhibitor’s CH_3_ symmetric and asymmetric stretch and aromatic ring CH stretch (Table A7, Table A8 and Table A9). Other similar compound incorporation contributions to the DPPG FTIR spectra are displayed and signalledin Figure 5b and correspond to characteristic inhibitor vibrational bands from the aromatic rings and functional groups. These comprise the signature-specific bands for each compound [56,57]. Overall, there is a verified increase in liposome band absorbance, with this being more preeminent in the DPPG methylene (2800–3000 cm^−1^), carbonyl (1736 cm^−1^), and phosphate (1224 and 1242 cm^−1^) groups (Figure 5).

According to the literature, the phosphate and carbonyl groups are considered hydration centers, with band position and strength providing useful information about the lipids’ hydration state when in contact with water [58,59,60]. Both regions are highly sensitive to local hydrogen bond interactions, and a decrease in hydrogen bonding (hydration state) results in a shift of phosphate and carbonyl bands to higher wavenumbers [59,60], thus providing great insight into the compounds’ bilayer membrane interaction mode. In Figure 5b, marked with a dashed line, and Table 7, it is perceived that the phosphate hydrogen-bonded band (1224 cm^−1^) presents a slight shift to higher wavenumbers around 1227 and 1228 cm^−1^ for Rucaparib and Veliparib, respectively, while Niraparib does not appear to significantly affect the phosphate hydrogen bond/hydration state.

DPPG carbonyl vibration modes can be evaluated by decomposition of the major band into three Gaussian peaks at 1699 cm^−1^, 1725 cm^−1^, and 1740 cm^−1^ (Figure A4). The first two bands are assigned to hydrogen-bonded carbonyl groups and the third band to non-hydrogen-bonded carbonyl groups, respectively [60]. Differences are perceived in the carbonyl peak profile upon inhibitor incorporation. A small shift to lower wavelengths is observed as a result of alterations in the Gaussian areas and contributions (Figure A4 and Table 8). An overall increase in the area of the non-hydrogen-bonded band (1740 cm^−1^) is observed, with the inhibitor impact following the tendency: Veliparib > Rucaparib > Niraparib (Table 8). Veliparib membrane incorporation appears to have a more profound effect with the surfacing of an additional vibrational band at 1760 cm^−1^ (Figure A4). More interestingly, it was verified that there was a significant alteration in the hydrogen-bonded bands, with a significant increase in the Gaussian area for the first peak and a decrease for the second peak (Figure A4), which is better perceived in the area ratio analysis in Table 8. Inhibitor impacts follow the same tendency as the third peak. Analysis from the area ratio values of the peak at 1725 cm^−1^ depicts the effect of each compound in the liposome bilayer, with values of 0.373, 0.035, 0.038, and 0.340 for the control, Veliparib, Rucaparib, and Niraparib, respectively (Table 8). 

The two low-frequency peaks for hydrogen-bonded carbonyls are attributed to different interaction modes, with the peak at 1699 cm^−1^ being associated with the presence of lipids’ glycerol OH in the interaction [60] and the peak at 1725 cm^−1^ being attributed to water OH molecules [58,60]. The major observed decrease in the second peaks’ area ratios indicates that the three compounds promote water displacement from the lipid membrane, while increasing another type of hydrogen-bond vibrational mode at lower wavelengths (1699 cm^−1^); see Figure A4 and Table 8. The increase in area and ratios of carbonyl’s first Gaussian (1699 cm^−1^) indicates a possible mode of interaction between all three compounds with the lipid membrane. The membrane insertion of PARP1 inhibitors may be contributing to the formation of additional hydrogen bond interactions with the lipids’ carbonyl groups, which may be orchestrated by the inhibitors’ amine groups/aromatic rings (that will be in a higher protonated state due to the solvent pH). This amine–carbonyl interaction, in turn, promotes an apparent membrane hydration that is, in fact, a water substitution. Confirmation of this event is further anchored by the narrowing of the X-H stretch (>3000 cm^−1^), where DPPG IR absorbance in the O-H region is decreased (>3440) and C-H and N-H are increased. 

Even though Niraparib appears to have the least impact on DPPG infrared spectra and membrane dehydration, which could be related to encapsulation efficiency, it appears that the major interaction and effect are directed to the carbonyl group. Regarding Veliparib and Rucaparib, membrane dehydration is favored by both compounds in the phosphate and carbonyl centers; however, a significant impact is verified in the liposome carbonyl group. To summarize, PARP1 inhibitors interact with the DPPG lipid membrane, preferentially in the carbonyl group, through hydrogen bonding. This interaction may involve the inhibitors’ protonated amine groups, which, in turn, leads to water displacement in the hydration centers.

To summarize, PARP1 inhibitor encapsulation was achieved in DPPG liposomes, and encapsulation efficiency was optimized taking into consideration DMSO content and its presence in a biocompatible range (below 0.05%). It was also observed that lipid–inhibitor interaction occurs preferentially in the carbonyl groups, thus confirming drug encapsulation in the lipidic membrane (Figure 6). The encapsulating DPPG formulation presented structural stabilization until 14 days of storage, which can be further increased by overall lipid formulation optimization (lipid conjugation) and liposome surface functionalization/coating. This work has shown that the devised DPPG + PARP1 inhibitor formulations present the best starting point and foundation for future target delivery design. By coating the surface of liposomes with polymers, polyelectrolytes, antibodies, and other compounds greater stabilization, increased time circulation, increased control release, and drug target delivery may be achieved.

## 4. Conclusions

In this work, lipid nano-delivery systems for three PARP1 inhibitors (Veliparib, Rucaparib, and Niraparib) were implemented and characterized. Simple lipid and dual lipid formulations with DPPG and DPPC were developed and tested following the thin-film method. Conditions were optimized for inhibitor encapsulation, where good values of encapsulation efficiency were attained (>40%), which translate to drug concentration values as high as 100 µM. DPPG-encapsulating inhibitors presented the best fit in terms of particle stabilization by retrieval of zeta potential values below −30 mV and good particle population distribution (single population profile). The size of the main population of interest was ~130 nm in diameter. DPPG/DPPC (1:1) formulations presented zeta potential values above −30 mV, indicative of particle destabilization, thus making DPPG liposomes the best formulation for PARP1 inhibitor encapsulation. An evaluation of the storage stabilization of DPPG formulations led to the verification of particle size stabilization until 14 days of storage at 4 °C. No particle deposition or flocculation was observed.

Kinetic release studies showed that DPPG-encapsulating inhibitors presented slower release rates than drug control samples, and DPPG + Veliparib/Niraparib were considered the most suitable formulations, with complete drug release delayed until 7 h. The Rucaparib formulation was efficient until 3 h of dialysis; however, after this timepoint, cumulative values converged to those of the control sample. In all three DPPG drug formulations, complex release mechanisms were identified, with Veliparib and Niraparib being governed by a combination of diffusion-controlled (Fickian) and non-Fickian diffusion, while anomalous and super case II transport was verified for Rucaparib, indicating that further optimization should be carried out to enhance system stabilization in circulation and drug release rates. This can be achieved using liposome membrane stabilization components such as cholesterol, PEG, polyelectrolytes, and other lipidic components.

Spectroscopic analysis revealed that PARP1 inhibitors interact with the DPPG lipid membrane, which may explain the complex release mechanism unveiled. Infrared analysis indicated a preferential membrane interaction with the lipid carbonyl groups through hydrogen bonding, where the inhibitors’ protonated amine groups may be major players in the drug encapsulation mode. These interactions appear to have a profound effect on lipid membrane hydration, where efficient PARP1 inhibitor encapsulation results in water displacement from phosphate and carbonyl hydration centers. Differences in the impact on membrane water displacement are perceived, with a tendency to follow the order from highest to lowest: Veliparib > Rucaparib > Niraparib. This could be explained by the encapsulation efficiencies of each drug, their molecular structure, and their composition. 

To summarize, stable single-lipid formulations for PARP1 inhibitors with high encapsulation values have been developed for therapeutic drug delivery. The type of interaction mode between the drug and lipid in the encapsulation process was determined, and an impact tendency among compounds was verified. Further optimization is needed to increase liposome membrane stabilization and optimize release rates and mechanisms in circulation. 

## Figures and Tables

**Figure 1 nanomaterials-13-01613-f001:**
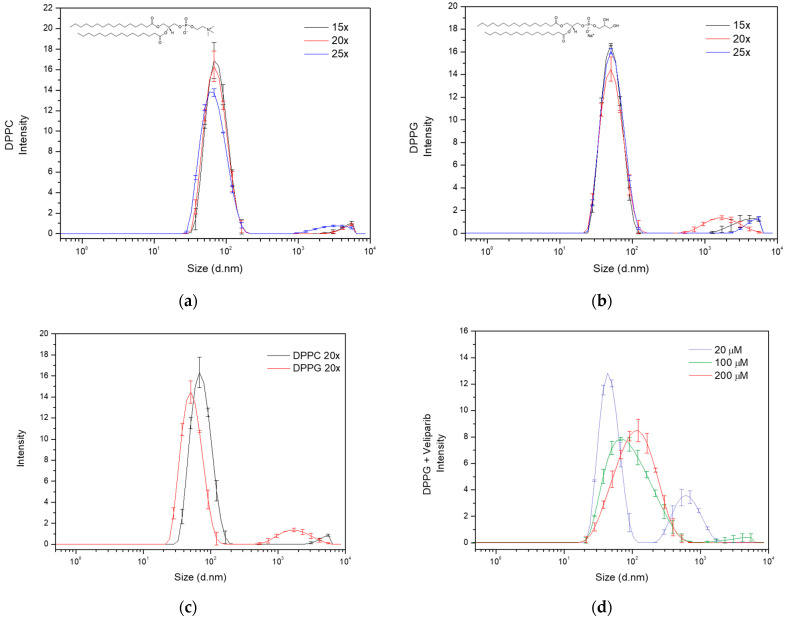
Size distribution analysis of liposome formulations dependent on sonication cycles and drug encapsulation. Evaluation of the effect of sonication cycles (15×, 20× and 25×) on (**a**) DPPC and (**b**) DPPG liposome particle size distribution. Analysis was carried out by determining the intensity of each size population detected by dynamic light scattering (DLS), and particle diameter values are presented in nanometers. The lipids’ structures are included in the respective graphs. The line in black refers to 15 repeats, the red line to 20 repeats, and the blue line to 25 repeats. (**c**) Population and size distribution of DPPC (dark line) and DPPG (red line) liposome formulations, produced with 20 sonication cycles (20×). Observation of bimodal distribution in both liposomes, with DPPG presenting the main population smaller than DPPC. The large size population presented diameter values above 500 nm in both formulations. (**d**) DPPG size distribution analysis dependent on supplemented Veliparib’s concentration in the liposome production.

**Figure 2 nanomaterials-13-01613-f002:**
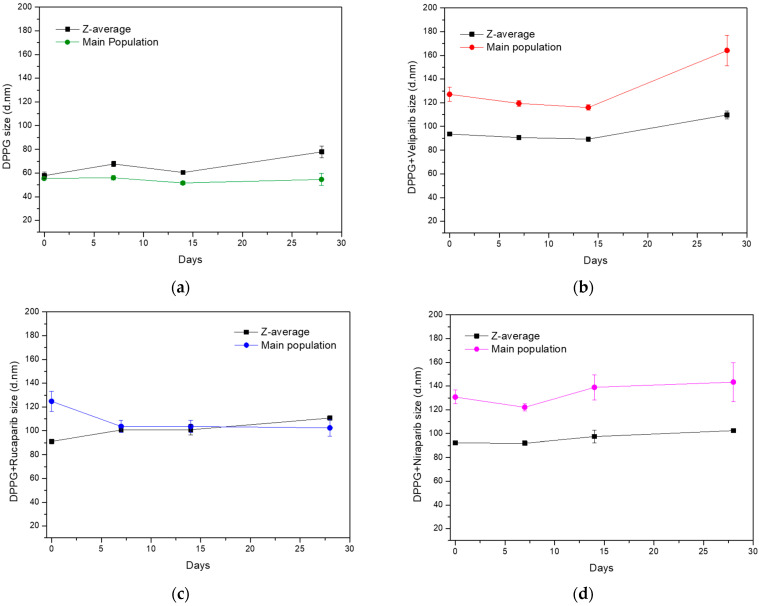
Evaluation of particle size stability (z-average and main population size values) of DPPG-encapsulating PARP1 inhibitors, during 28 days of storage at 4 °C. (**a**) Control sample without an inhibitor, (**b**) DPPG with Veliparib, (**c**) DPPG with Rucaparib, and (**d**) DPPG with Niraparib. In all graphs, z-average (black line) and main population size values (colored line) are presented as mean values with standard deviation values (SD).

**Figure 3 nanomaterials-13-01613-f003:**
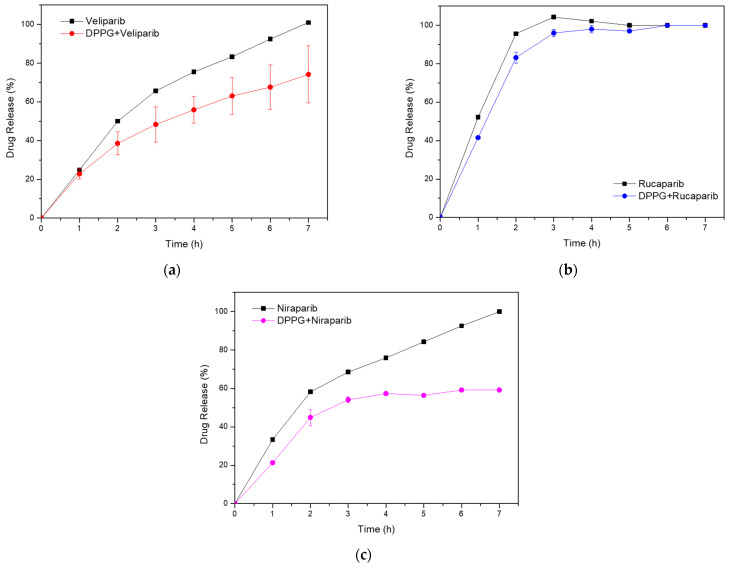
In vitro cumulative drug release analysis from DPPG liposomes (colored line), with respective drug control samples (black line). (**a**) Veliparib; (**b**) Rucaparib; (**c**) Niraparib.

**Figure 4 nanomaterials-13-01613-f004:**
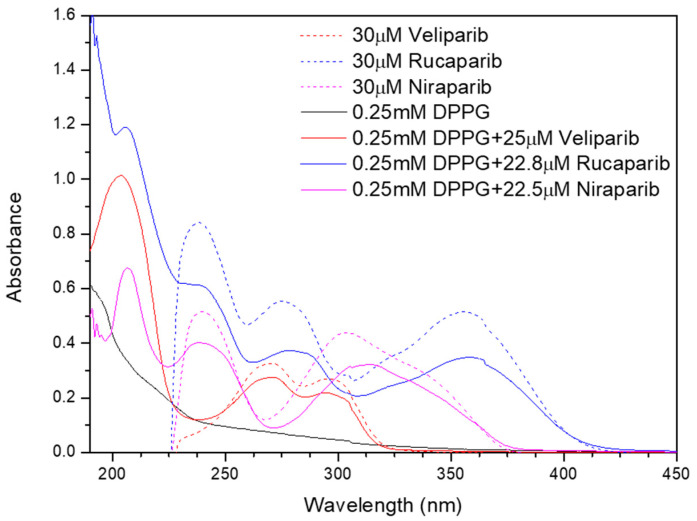
Absorption spectra of DPPG (black line), DPPG+PARP1 inhibitor liposomes (colored line), and pure inhibitor solutions (colored dash line).

**Figure 5 nanomaterials-13-01613-f005:**
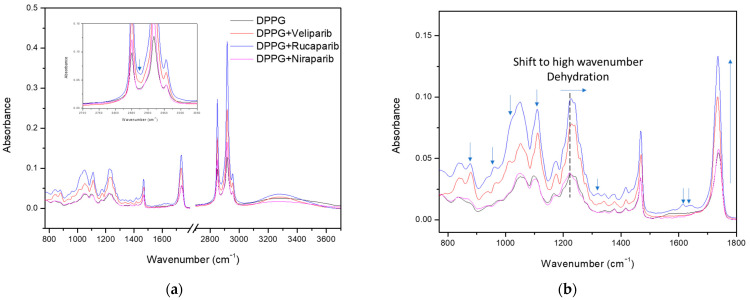
FTIR analysis of DPPG liposomes with and without PARP1 inhibitors. (**a**) Full spectra with the inset showing the presence of the inhibitor band in the 2800–3000 cm^−1^ region. (**b**) Enlarged spectra, displaying the differences between the DPPG control and encapsulating samples. Identification of the major contributions to DPPG spectra changes due to inhibitor encapsulation in the lipid membrane. Arrows identifying the inhibitor contribution bands, absorbance increases in the carbonyl and phosphate bands, and the phosphate band at 1224 cm^−1^ shift toward higher values (indication of possible lipid membrane dehydration in the inhibitor encapsulating formulae).

**Figure 6 nanomaterials-13-01613-f006:**
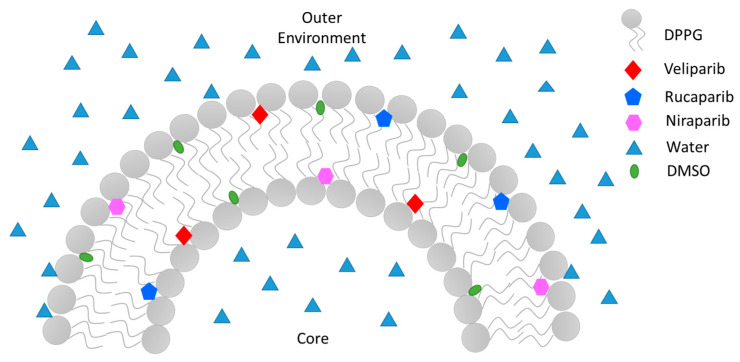
Illustration of PARP1 inhibitors’ Veliparib, Rucaparib, and Niraparib encapsulation mode in a sectionized DPPG liposome’s lipid bilayer.

**Table 1 nanomaterials-13-01613-t001:** Nanoparticle characterization of DPPC and DPPG liposome formulations produced by 20 sonication cycles (20×). Diameter values for each population, zeta average, polydispersity index, and zeta potential values are presented.

Lipid Formulation *	Z-Average (nm)	PdI	Pop.1 (nm)	Pop.2 (nm)	Zeta Potential (mV)
DPPC (20×)	71.0 ± 0.5	0.18 ± 0.01	75 ± 2	4700 ± 200	−17.9 ± 0.9
DPPG (20×)	56.2 ± 0.4	0.27 ± 0.01	56 ± 2	1900 ± 200	−43 ± 2 mV

* Mean ± SD values. Note: PdI—polydispersity index, Pop.1—population 1, and Pop.2—population 2.

**Table 2 nanomaterials-13-01613-t002:** Veliparib encapsulation efficiency (EE) and lipid capacity (LC) values in DPPG liposomes.

Veliparib (µM)	EE * (%)	LC * (%)
20	99.9 ± 0.2	1.99 ± 0.01
100	49± 2	3.9 ± 0.2
200	55 ± 1	10.20 ± 0.09

* Mean ± SD values.

**Table 3 nanomaterials-13-01613-t003:** DPPG liposomes’ particle size and zeta potential variance dependent on Veliparib concentration.

Veliparib (µM)	Particle Size *				Zeta Potential * (mV)
	Z-Average (d.nm)	PdI	Pop.1 (d.nm)	Pop.2 (d.nm)	
20	60 ± 2	0.40 ± 0.04	48 ± 1	741 ± 86	−31 ± 2
100	79.4 ± 0.4	0.29 ± 0.02	116 ± 7	~2700	−41 ± 2
200	91 ± 2	0.28 ± 0.03	134 ± 6	---	−39 ± 2

* Mean ± SD values. Note: PdI—polydispersity index, Pop.1—population 1, and Pop.2—population 2.

**Table 4 nanomaterials-13-01613-t004:** Encapsulation efficiency (EE) and lipid capacity (LC) values of PARP1 inhibitors in single- and dual-lipid formulations.

PARP1Inhibitor	MolecularStructure	DPPG(200 µM) *	DPPG/DPPC 1:1(100 µM) *
EE ^+^ (%)	LC ^+^ (%)	EE ^+^ (%)	LC ^+^ (%)
Veliparib	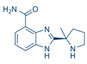	55 ± 1	10.20 ± 0.09	38 ± 2	3.3 ± 0.3
Rucaparib	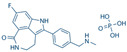	49.2 ± 0.3	9.2 ± 0.1	40 ± 1	2.80 ± 0.03
Niraparib	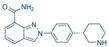	50.7 ± 0.9	9.0 ± 0.1	33.7 ± 0.9	3.1 ± 0.2

* Supplemented drug concentration; ^+^ mean ± SD values.

**Table 5 nanomaterials-13-01613-t005:** DLS particle size and zeta potential characterization of DPPG and PG/PC (1:1) formulations encapsulating PARP1 inhibitors.

Liposome	PARP1 Inhibitor	Particle Size *				Zeta Potential * (mV)
		Z-Average (d.nm)	PdI	Pop.1 (d.nm)	Pop.2 (d.nm)	
DPPG	Veliparib	91 ± 2	0.28 ± 0.02	130 ± 3	----	−39 ± 2
	Rucaparib	94 ± 8	0.25 ± 0.01	135 ± 10	----	−33 ± 1
	Niraparib	94 ± 4	0.25 ± 0.01	132 ± 7	----	−36 ± 2
DPPG/DPPC 1:1	Veliparib	88 ± 2	0.22 ± 0.01	115 ± 4	----	−29 ± 4
	Rucaparib	97 ± 3	0.28 ± 0.02	130 ± 40	~117	−25 ± 4
	Niraparib	101 ± 2	0.31 ± 0.05	156 ± 11	----	−25 ± 4

* Mean ± SD values. Note: PdI—polydispersity index, Pop.1—population 1, and Pop.2—population 2.

**Table 6 nanomaterials-13-01613-t006:** Release kinetics determined for PARP1 inhibitors from DPPG liposomes.

Model	Veliparib				Rucaparib				Niraparib			
	R^2^	K	β	AIC	R^2^	K	β	AIC	R^2^	K	β	AIC
Weibull	0.999	4.33	0.94	31.76	0.992	0.75	1.02	60.29	0.999	4.63	0.93	54.15
Weibull (*T_LAG_*)	0.999	3.78	0.83	16.64	0.989	0.75	1.02	62.29	0.999	3.59	0.73	50.60
	R^2^	Km		AIC	R^2^	Km		AIC	R^2^	Km		AIC
Michaelis–Menten	1.00	0.55		60.89	1.00	1.15		54.16	1.00	0.54		55.53
Michaelis–Menten (*T_LAG_*)	1.00	0.55		62.92	1.00	1.13		56.22	1.00	0.53		57.56
	R^2^	Km		AIC	R^2^	Km		AIC	R^2^	Km		AIC
Hixson–Crowell	0.629	0.45		67.81	0.512	0.47		78.78	0.537	0.40		69.72
	R^2^	n		AIC	R^2^	n		AIC	R^2^	n		AIC
Korsmeyer–Peppas	0.999	0.92		58.91	0.998	0.94		76.55	0.998	0.92		65.36
Korsmeyer–Pepas (*T_LAG_*)	0.999	0.72		47.65	0.999	0.73		72.48	0.999	0.72		60.68

Note: R^2^ refers empirical coefficient of determination; AIC—Akaike information criterion; K—model release constants; β—shape parameter; Km—Michaelis–Menten constant; n—diffusion release exponent.

**Table 7 nanomaterials-13-01613-t007:** Analysis of the hydrogen-bonded phosphate band (1224 cm^−1^) shift dependent on inhibitor encapsulation in DPPG liposomes. Arrows indicate the direction of wavenumber band shift in the FTIR spectra upon inhibitor encapsulation.

Hydrogen-Bonded Phosphate Band (1224 cm^−1^) Shift			
DPPG	DPPG + Veliparib	DPPG + Rucaparib	DPPG + Niraparib
1224 (0)	1227.7 (+3.7)	1226.5 (+2.5)	1223.4 (−0.6)
-	→	→	←

**Table 8 nanomaterials-13-01613-t008:** Area ratios of carbonyl groups deconvoluted with Gaussian curves, corresponding to hydrogen-bonded (maximum band at 1699 cm^−1^ and 1725 cm^−1^ wavenumbers) and non-hydrogen-bonded forms (maximum band at 1740 cm^−1^ wavenumbers).

DPPG Formulation	Ratios (A _x cm_^−1^/A _total_)			
	A _1699 cm_^−1^	A _1725 cm_^−1^	A _1699+1725 cm_^−1^	A _1740 cm_^−1^
Control	0.066	0.373	0.439	0.560
Veliparib	0.179	0.035	0.214	0.774
Rucaparib	0.170	0.038	0.208	0.792
Niraparib	0.074	0.340	0.414	0.586

## Data Availability

Data are contained within the article. The data presented in this study are available in the [insert article and Appendix A here].

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
