# Peer review of "Liposome Formulations for the Strategic Delivery of PARP1 Inhibitors: Development and Optimization"

_nanomaterials, 2023, doi:10.3390/nano13101613_

Round 1
Reviewer 1 Report
Manuscript of Conceição and co-authors describes lipid nano-delivery systems for three PARP1 inhibitors including preparation, characterization and analysis of drug release. The lipidic nano-particles are well characterized. The type of interaction mode between drug and lipid in the encapsulation process was determined and an impact tendency among compounds was verified. The manuscript is well written and results are clear presented. I believe that this manuscript is of interest for readers of Nanomaterials and has potential for publication after minor revisions listed below.
- Part 3.4. Spectroscopic characterization. UV-Vis spectra of pure inhibitors should be added for comparison. Also, is there any contribution from light scattering? What conclusion do the authors draw from the UV-Vis spectra?
- From the analysis of FTIR spectra (Fig. 7), the authors conclude that the shift of the phosphate band vibration bands (1224 cm-1, Fig. 7b) is a sign of membrane dehydration. However, I do not clearly see such a shift. Additional vibration band analysis and explanations are required.
The quality of the English in the submitted text meets the requirements of the journal
Author Response
Response to the Comments
We thank the reviewer for the comments and suggestions that we think are properly addressed in the present form of the manuscript. We believe that corrections made fully accomplished reviewers comments and suggestions and improved its overall quality has been achieved
Reviewer1:
Manuscript of Conceição and co-authors describes lipid nano-delivery systems for three PARP1 inhibitors including preparation, characterization and analysis of drug release. The lipidic nano-particles are well characterized. The type of interaction mode between drug and lipid in the encapsulation process was determined and an impact tendency among compounds was verified. The manuscript is well written and results are clear presented. I believe that this manuscript is of interest for readers of Nanomaterials and has potential for publication after minor revisions listed below.
- Part 3.4. Spectroscopic characterization. UV-Vis spectra of pure inhibitors should be added for comparison. Also, is there any contribution from light scattering? What conclusion do the authors draw from the UV-Vis spectra?
Answer from Authors: We thank the reviewer´s comments and we have included the UV-vis spectra for the pure drug solutions in the now figure 4 in section 3.4. In this study, the main goal was to attain the encapsulation of specific PARP1’s inhibitors in the liposomes and verify if any type of structural change would occur to either of the inhibitors or liposomes due to the procedure. By analysis of the UV-vis spectra of encapsulated drug shown in section 3.4,ex- Fig. 6, now Fig.4, no modification was detected, thus confirming that no loss of structural integrity occurs during the liposome production protocol. Also, no indication of lipid oxidation due to production procedure was observed in the liposomes’ UV spectra. Additionally, we have confirmed encapsulation by detection of specific inhibitors’ absorbance peaks in higher wavenumbers than the typical lipid absorbance region. Regarding light scattering events, the spectra form pure drug solution did not present light scattering, nor the liposome formulation spectra with and without inhibitor. To prevent the possibility of light scattering, all liposome samples analysed were diluted in the moment of UV-vis analysis. The dilution factor was taken into consideration for determination of drug concentration and subsequent EE and LC determination.
- From the analysis of FTIR spectra (Fig. 7), the authors conclude that the shift of the phosphate band vibration bands (1224 cm-1, Fig. 7b) is a sign of membrane dehydration. However, I do not clearly see such a shift. Additional vibration band analysis and explanations are required.
Answer from Authors: We thank this comment. In comparison to the carbonyl groups, the phosphate band shift is smaller. Due to the difficulty on visualizing phosphate band shift in the graph we have added a table reporting the values of the peak positions of hydrogen-bonded phosphate band as well as respective shift in relation to DPPG the measured on the formulations in section 3.4.
Reviewer 2 Report
Dear Authors,
I recommend unloading the text by transferring some tables and figures to the Supplementary. In particular, tables 9-11.
Sincerely yours, Reviewer
Author Response
Response to the Comments
We thank the reviewer for the comments and suggestions that we think are properly addressed in the present form of the manuscript. We believe that corrections made fully accomplished reviewers comments and suggestions and improved its overall quality has been achieved.
Reviewer 2:
I recommend unloading the text by transferring some tables and figures to the Supplementary. In particular, tables 9-11.
Answer from Authors: We thanks the comment. Table 7 was transferred to appendix A as tables 9-11.
Reviewer 3 Report
1. Please avoid using abbreviations in the abstract. For example, PARP1, DPPG and DPPC.
2. The keywords “PARP1: inhibitors” should be “PARP1 inhibitors” and “DPPG” should be removed or replaced with “lipid formulation”.
3. All the purchase details of chemicals/reagents and instruments/equipment/software/kits should be provided as state, city, and country in the case of the USA as well as city and country in the case of other countries. Also, for the second instance of the same vendor/company’s mention, the authors can simply mention the company name, for instance, Sigma-Aldrich and not every time Sigma-Aldrich (USA).
4. A reference should be provided for the section 2.2 and 2.5.
5. There is a mixed usage of abbreviations such as C20x and G20x, PC and PG as well as DPPC and DPPG. Be consistent with one abbreviation like DPPC and DPPG throughout the manuscript including tables and figures.
6. The PARP1 inhibitors should be abbreviated as V, R and N for Veliparib, Rucaparib and Niraparib respectively, instead of mixed used of full form in some places and abbreviated form (V, R and N) in some places.
7. All the abbreviations used in the tables should be explained in full form in the respective footers and likewise that used in Figures in the respective captions.
8. As too elaborate discussion is done, the section 3 should be reduced in length and so do the conclusion to make them more clear and concise.
9. Some recent references are missing such as https://doi.org/10.3390/app122312347; https://doi.org/10.1016/j.foodchem.2021.130611; https://doi.org/10.3390/nu11051052.
Minor editing of English language required
Author Response
Response to the Comments
We thank the reviewer for the comments and suggestions that we think are properly addressed in the present form of the manuscript. We believe that corrections made fully accomplished reviewers comments and suggestions and improved its overall quality has been achieved.
1.Please avoid using abbreviations in the abstract. For example, PARP1, DPPG and DPPC.
Answer from Authors: We thank this suggestion. The full name for each of the molecules was added in the abstract.
2.The keywords “PARP1: inhibitors” should be “PARP1 inhibitors” and “DPPG” should be removed or replaced with “lipid formulation”.
Answer from Authors: The issue was addressed as suggested by the reviewer.
3.All the purchase details of chemicals/reagents and instruments/equipment/software/kits should be provided as state, city, and country in the case of the USA as well as city and country in the case of other countries. Also, for the second instance of the same vendor/company’s mention, the authors can simply mention the company name, for instance, Sigma-Aldrich and not every time Sigma-Aldrich (USA).
Answer from Authors: The complementation of address information following the suggestions of the reviewer was carried out.
4.A reference should be provided for the section 2.2 and 2.5.
Answer from Authors: References were added into Sections 2.2 had 2.5.
5.There is a mixed usage of abbreviations such as C20x and G20x, PC and PG as well as DPPC and DPPG. Be consistent with one abbreviation like DPPC and DPPG throughout the manuscript including tables and figures.
Answer from Authors: We thank the revisor for the correction and the issue was addressed in the complete extension of the manuscript.
6.The PARP1 inhibitors should be abbreviated as V, R and N for Veliparib, Rucaparib and Niraparib respectively, instead of mixed used of full form in some places and abbreviated form (V, R and N) in some places.
Answer from Authors: We thank this comment and we decided to use the full name when referencing the compounds along the manuscript. The uniformization was carried out along all manuscript including figures and tables.
.
7.All the abbreviations used in the tables should be explained in full form in the respective footers and likewise that used in Figures in the respective captions.
Answer from Authors: The issue was addressed as suggested.
8.As too elaborate discussion is done, the section 3 should be reduced in length and so do the conclusion to make them more clear and concise.
Answer from Authors: We reduced a bit the size of section 3 by combining of Fig. 1, 2 and 3.
9.Some recent references are missing such as https://doi.org/10.3390/app122312347; https://doi.org/10.1016/j.foodchem.2021.130611; https://doi.org/10.3390/nu11051052.
Answer from Authors: Thank you for this suggestion. These references were added in section 1.
Reviewer 4 Report
The manuscript presents a detailed experimental study of the encapsulation of three different PARP1 inhibitors inside giant vesicles. The authors describe the methods used for encapsulation, as well as the trapping efficiency. The method used, based on the application of high-energy ultrasound, gives two population of vesicles, one in the nanometric and the other in the micrometer scales. The delivery rate of the inhibitors is measured by spectroscopic methods. The systems are stable for a maximum or 14 days, when the delivering capsules aggregate and phase separate. The content is suitable for the journal, however, it needs revision before it can be accepted.
COMMENTS:
1.- The authors have to define the meaning of each acronym before it appears for the first time.
2.- The paper discusses the encapsulation in only one type of capsules, liposomes and vesicles. Would it be more or less efficient to encapsulate the PARP1 molecules in other, such as lipid nanoparticles coated with other biomolecules (see Ruano et al.)?
3.- In order to be efficient from the medical point of view, the capsules should deliver the cargo molecules specifically to the tumoral cells. This objective cannot be reached with the present delivery platforms.
4.- Which is the effect of the DMSO remaining in the capsules in their biocompatibility? The amount of DMSO in the membranes has not been quantified. Do the authors figure out if there any way to estimate separately the efficiency of the PARP1 inhibitors and the remaining DMSO?
5.- Page 3, third line of section 2.4: The measuring angle "theta" has not been defined.
6.- Section 2.4.- Whis is the temperature used during the DLS and theta potential measurements?
7.- Section 2.4: The dispersant is supposed to be water, that has a different refractive index, otherwise DLS measurements would not be possible. Please discuss this.
8.- Section 2.5: Indicate that measurements of release of inhibitors have been done at 37ºC because it is the human body temperature. Since the transition temperature of one of the phospholipids is 41ºC, and there is DMSO in the membrane, it is expected that the transition temperature decreases, which would strongly affect the delivery rate. This has to be discussed, and the transition temperature measured.
It is fine
Author Response
Response to the Comments
We thank the reviewers for the comments and suggestions that we think are properly addressed in the present form of the manuscript. We believe that corrections made fully accomplished reviewers comments and suggestions and improved its overall quality has been achieved.
1. The manuscript presents a detailed experimental study of the encapsulation of three different PARP1 inhibitors inside giant vesicles. The authors describe the methods used for encapsulation, as well as the trapping efficiency. The method used, based on the application of high-energy ultrasound, gives two population of vesicles, one in the nanometric and the other in the micrometer scales. The delivery rate of the inhibitors is measured by spectroscopic methods. The systems are stable for a maximum or 14 days, when the delivering capsules aggregate and phase separate. The content is suitable for the journal, however, it needs revision before it can be accepted.
- The authors have to define the meaning of each acronym before it appears for the first time.
Answer from Authors: We thank the reviewer and state that the issue was addressed in the manuscript as requested.
- The paper discusses the encapsulation in only one type of capsules, liposomes and vesicles. Would it be more or less efficient to encapsulate the PARP1 molecules in other, such as lipid nanoparticles coated with other biomolecules (see Ruano et al.)?
Answer from Authors: Thank you very much for this comment which is in fact relevant for the development of PARP1 inhibitors carriers. We believe that the encapsulation of PARP1 inhibitors is strongly dependent on the lipid, or the other molecules used, overall charge and also the interact location between lipid and coat polymers that are used. In this work, our goal was verifying which type of lipids are advised to increase drug encapsulation since these PARP1 inhibitors are molecules liposoluble. In the next stage of our research, one intend use both polyelectrolytes and lipids adsorbed on magnetic nanoparticle since it was already demonstrated that both lipidic bilayers and liposomes can be adsorbed on polyelectrolyte layers achieved by the layer-by-layer technique (https://doi.org/10.1017/S1431927613001621 and https://doi.org/10.1017/S1431927619000345 ). Therefore, our goal will be in accordance with the Ruando et al. (https://doi.org/10.1021/acs.langmuir.1c00341) work, butin our case starting with magnetic nanoparticles covering with polyelectrolyte layers, lipidic bilayers carrying the PARP1 inhibitors and if necessary with other biological molecules. In accordance with our results, one expects if DOPC:DODAB is used, as used, then the encapsulation of the studied inhibitors should have an extremely low or even no drug encapsulation, solely by analysing lipid structure and overall zeta-potential values. Therefore, we demonstrated that the lipids used allow high encapsulation efficiency. with the possibility for further application of surface coatings with other molecules such as polymers, polyelectrolytes, antibodies, etc and/or in conjugation with other typed of scaffolds. Generally, we demonstrated that PARP1 inhibitors encapsulation is favoured using extremely negatively charged lipids and that DPPG should be employed when designing a lipid carrier for these compounds. Additionally, attention should be taken when choosing the best combination between drug, lipid and polyelectrolyte/polymer coating.
Taking into consideration this comment, we have reinforced the applicability here discussed and added this information in the results and discussion section (section 3.2 and section 1).
- In order to be efficient from the medical point of view, the capsules should deliver the cargo molecules specifically to the tumoral cells. This objective cannot be reached with the present delivery platforms.
Answer from Authors: We agree with the reviewer. As the present work is the first stage of fundamental research to understand how PARP1 inhibitors, specifically the ones that are usually developed by mimicking NAD+ molecules, as the ones in the study, interact with two potential liposome formulations and how these simpler formulations would behave. We have tried to optimize the encapsulation efficiency of these compounds since the DMSO solvent is a great limitation for further drug encapsulation and for the application of these systems in any type of treatment, since the usage of DMSO should be limited to a biocompatible percentage. So, we think we have helped to uncover it in this work and will be helpful for other researchers. We understand that these formulations are still not a full-fledge target delivery liposome system, and we state that these formulations need further optimization to increase stabilization. However, the system DPPG-inhibitor developed presents the best start point as foundation for future target design and liposome system complexation to convey greater stabilization, time circulation, increase control release and drug target delivery, which could be achieved by addition of molecules, such as polymers, polyelectrolytes, antibodies and among others, to liposomes’ surface thus achieving specific delivery to tumor masses/cells.
Taking into consideration this comment, we have explained this further in the manuscript in section 3.4 and conclusions.
4.- Which is the effect of the DMSO remaining in the capsules in their biocompatibility? The amount of DMSO in the membranes has not been quantified. Do the authors figure out if there any way to estimate separately the efficiency of the PARP1 inhibitors and the remaining DMSO?
Answer from Authors: Thank you for this important comment. In this work we have optimized a method that helps to incorporate as much inhibitor as possible, taking into consideration the DMSO limitation, both in terms of drug/DMSO competition for the lipid’s headspace, but also in terms of biocompatibility for future therapeutic application. As such, we decided upon drug addition to use drug concentrations that would input on the system DMSO concentrations far below the 0.1 – 0.5% (v/v), which is the limit advised to use in in-vitro cell culture (doi: 10.3390/molecules22111789). When employing the 100 µM and 200 µM inhibitor supplementation to the liposomes effective % DMSO was of 0.1% and 0.2% respectively. But we have verified that after dialysis the DMSO percentage was decreased significantly by the observation of a decrease in absorbance at ~205 nm, wavelength associated to typical DMSO maximum absorbance peak region in the liposomes’ UV-vis spectra. From our analysis we have determined that after dialysis the final DPPG formulations present DMSO at 0.02% and 0.05% for the 100 µM and 200 µM inhibitor supplementation, respectively, which are within the 0.05% DMSO limit value for safe application in almost all cells and determined by Gallardo-Villagran et al., 2022 for Human Fibroblast-like Synoviocytes (doi: 10.3390/molecules27144472). Furthermore, since the 0.02-0.05% DMSO content refers to the liposome stock solution, through the dilution for cell testing the effective DMSO content will be further decreased on culture medium and/or organism of study.
Taking into consideration this comment, we have added this information in section 3.1.2.
- Page 3, third line of section 2.4: The measuring angle "theta" has not been defined.
Answer from Authors: The measuring angle was already stated in the 3rd line in parentheses, but we have reinforced the information in section 2.4.
- Section 2.4.- Whis is the temperature used during the DLS and theta potential measurements?
Answer from Authors: The temperature defined and used during the measurements was 25⁰C, which we added upon the reviewer comment.
- Section 2.4: The dispersant is supposed to be water, that has a different refractive index, otherwise DLS measurements would not be possible. Please discuss this.
Answer from Authors: The authors apologize this was a typing mistake during the manuscript elaboration, the refractive index for liposome used was 1.45 and not 1.33. This value provided by the lipids manufacturer. This value was corrected in the manuscript.
- Section 2.5: Indicate that measurements of release of inhibitors have been done at 37ºC because it is the human body temperature. Since the transition temperature of one of the phospholipids is 41ºC, and there is DMSO in the membrane, it is expected that the transition temperature decreases, which would strongly affect the delivery rate. This has to be discussed, and the transition temperature measured.
Answer from Authors: Thank you very much for this important question. The information for temperature was added in the section 2.5.
In regard to the effect of DMSO in phospholipids transition temperature, we have discussed this in the section 3.1.2, 5-6th paragraph and we have mentioned previous reports that state that DMSO at molar fractions below 0.1,which is within the region that we worked – values below 5x10-4), and do not appear to change membrane structure and headgroup mobility, even though water displacement is verified by competition and wining of lipids headgroups. At these ratios lipids’ gel-to-fluid transition temperature increases, stabilizing gel phase by changes in the surface hydration on lipid head-group and thermostability can be extended by DMSO up to a 5⁰C increase (doi:10.1016/j.bpj.2015.06.011 and doi:10.1016/j.bpj.2020.05.037). Because DMSO affects in a positive feedback lipids transition temperature, even at an incubation of 37ºC that is lower than Tm, the liposome formulations should be able to sustain the condition employed in the assay and delay drug release which we have proved in our drug release assays present in section 3.3. What we verified was that the liposome-drug formulations, at least for Veliparib and Niraparib, presented a complex drug release mechanism that is associated to combined mechanisms of diffusion and controlled transport by nanoparticle swelling and deswelling events. Only DPPG-Rucaparib present anomalous and super case II transport that is associated to membrane swelling events and presence of porous structures in the membrane that can be resultant from the DMSO presence but also due to the presence of phosphate counterion from original drug formulation.
However, we do understand the importance and necessity of determining the effective Tm from the developed formulations and this is one of the tasks and objectives that we have included to do in future work.
Round 2
Reviewer 4 Report
The authors have included in the new version the answers to my comments to the original manuscript. Therefore, I consider that the new version can be accepted for publication.
Author Response
We thank you for the comments and suggestions..